# Altered Intracellular Signaling Associated with Dopamine D2 Receptor in the Prefrontal Cortex in Wistar Kyoto Rats

**DOI:** 10.3390/ijms24065941

**Published:** 2023-03-21

**Authors:** Agata Korlatowicz, Magdalena Kolasa, Paulina Pabian, Joanna Solich, Katarzyna Latocha, Marta Dziedzicka-Wasylewska, Agata Faron-Górecka

**Affiliations:** Department of Pharmacology, Maj Institute of Pharmacology, Polish Academy of Sciences, Smętna 12, 31-343 Kraków, Poland

**Keywords:** Wistar-Kyoto rats, dopamine D2 receptor, intracellular signaling, βarrestin2/AKT/Gsk-3β/β-catenin pathway

## Abstract

Wistar-Kyoto rats (WKY), compared to Wistar rats, are a well-validated animal model for drug-resistant depression. Thanks to this, they can provide information on the potential mechanisms of treatment-resistant depression. Since deep brain stimulation in the prefrontal cortex has been shown to produce rapid antidepressant effects in WKY rats, we focused our study on the prefrontal cortex. Using quantitative autoradiography, we observed a decrease in the binding of [^3^H] methylspiperone to the dopamine D2 receptor, specifically in that brain region—but not in the striatum, nor the nucleus accumbens—in WKY rats. Further, we focused our studies on the expression level of several components associated with canonical (G proteins), as well as non-canonical, D2-receptor-associated intracellular pathways (e.g., βarrestin2, glycogen synthase kinase 3 beta—*Gsk-3β*, and β-catenin). As a result, we observed an increase in the expression of mRNA encoding the regulator of G protein signaling 2-RGS2 protein, which is responsible, among other things, for internalizing the D2 dopamine receptor. The increase in RGS2 expression may therefore account for the decreased binding of the radioligand to the D2 receptor. In addition, WKY rats are characterized by the altered signaling of genes associated with the dopamine D2 receptor and the βarrestin2/AKT/Gsk-3β/β-catenin pathway, which may account for certain behavioral traits of this strain and for the treatment-resistant phenotype.

## 1. Introduction

The Wistar-Kyoto (WKY) strain was initially developed in 1971 as a normotensive control strain for the spontaneously hypertensive rat (SHR). WKY rats were derived from outbred Wistar (WIS) rats. When comparing groups of WKY to WIS animals, it has been shown that the WKY strain exhibits a high hypersensitivity to stress. As a result of numerous studies using this strain in the context of depression, WKY rats have been recognized as a genetic model of endogenous treatment-resistant depression (TRD) with several important similarities to the human condition [1,2,3].

The WKY rats display behavioral and neurobiological phenotypes that are similar to those observed in clinical cases of major depression. These animals are unable to initiate or regulate the stress response appropriately, which predisposes them to adopt passive coping strategies in a response to aversive stimuli. Such a maladaptive stress response, resulting from the relatively greater impact of stressful stimuli, makes this strain more susceptible to developing depressive- and anxiety-like phenotypes, which are present in the absence of exposure to explicit stress. The exact genetic and molecular mechanisms underlying the depressive-like phenotype of WKY rats are still unresolved. Overall, there is some evidence for a decreased reward sensitivity in WKY rats. It has also been shown that WKY rats were less responsive than the control strains to the rewarding effects of nicotine [4] or morphine [5], which were in place as the conditioning procedures. On the other hand, using a chronic mild stress procedure (CMS) [6] did not result in the detection of abnormalities in the sucrose consumption measure, either at baseline or following exposure to CMS. However, they were able to demonstrate that WKY rats did show evidence of increased anxious behavior in the elevated plus maze and novel object recognition tests, the ones relative to WIS rats, and a greater impact of CMS on body weight gain and open arm entries. In addition, WKY rats do not respond to antidepressant administration in animal models of depression, apart from ketamine (KET) and deep brain stimulation (DBS) [7]. Neurochemical and endocrine dysregulations, in comparison to WIS rats, were identified and well described in the recent review by Aleksandrova et al. (2019) [8]. Among others, there is strong evidence of abnormalities in monoamine systems (5-hydroxytryptamine–5HT, dopamine–DA and norepinephrine-NE). WKY rats exhibit a lower level of all three monoamines in various brain regions, as well as dysfunctions in the amine turnover system; there are also alterations in the level of various aminergic receptors (e.g., a lower level of beta-adrenergic receptors, a higher expression of postsynaptic 5-HT1A receptors, as well as a lower expression of somatostatin receptors type 2 and 4 (Sst2, Sst4) [9]).

Besides other monoaminergic systems, mesolimbic and mesocortical dopamine functions are decreased in WKY rats [10,11,12,13,14,15]. These functions may also contribute to depressive-like behaviors, including anhedonia, social withdrawal, and cognitive deficits. Such a notion is justified when considering an interesting hypothesis, which can explain this phenomenon, that suggests that the anhedonia observed in patients with depression is related to a decrease in the sensitivity of postsynaptic D2 dopamine receptors (D2R) in the nucleus accumbens that are responsible for the reward function, as well as those connecting the therapeutic effects of long-term antidepressant administration with the increase in sensitivity of D2R in these brain areas [16]. Recently, it was shown that dopamine D2R in the prefrontal cortex (PFc) is responsible for the action of venlafaxine (the antidepressant belonging to serotonin and noradrenaline reuptake inhibitors) in order to reverse the impairment of memory consolidation that is caused by chronic stress in WIS rats. Interestingly, D2R was not responsible for the effects of DBS in WKY rats [6]. Therefore, a question presents itself: Is the lack of transmission in D2R-related signaling responsible for drug resistance in WKY rats?

Dopamine receptor functions have typically been associated with the regulation of cyclic adenosine monophosphate-cAMP and protein kinase A (PKA) via G protein-mediated signaling. D2R is coupled to G alpha subunits (Galpha i/o) and negatively regulates the production of cAMP, resulting in the decrease in the PKA activity and influencing the level of its substrates (including dopamine- and cAMP-regulated neuronal phosphoprotein; DARPP-32), as well as other cAMP-regulated molecules (e.g., exchange proteins Epac 1 and Epac2). This picture has been further complicated and includes the interaction of the receptor with βarrestins, the activation of mitogen-activated protein kinase MAP kinases (ERK), the modulation of calcium channels and potassium channels (GIRK), as well as protein kinase C and its signaling [17].

Important mechanisms for the regulation of G-protein-mediated signal transduction involve the specific GTPase-activating proteins of the regulators of the G protein signaling (RGS) family [18]. This family of proteins includes at least 37 members that are characterized by the presence of the same 125-amino-acid sequence, the so-called RGS box or RH homology domain, which binds the GTP-bound G protein alpha subunits and dramatically accelerates the rate of GTP hydrolysis [19]. Thus, RGS proteins act as GTPase-accelerating proteins by facilitating the return of the G protein alpha subunits to the inactive state. By reducing the lifetimes of the G alpha–GTP and the beta/gamma-subunit complexes, the RGS proteins act as negative modulators of the G protein signaling and can affect both the potency and the efficacy of the agonist action and downstream signaling [20]. It has been shown that D2R signaling can be mediated by several RGS proteins, including RGS4, RGS7, RGS9, and RGS2 [21,22,23,24].

In the present work, we decided to examine the level of D2R in the prefrontal cortex (PFc) of WKY and WIS Han (WIS) rats, as well as some of the essential components of D2R signaling, including the subtypes of RGS and RGS2, which are associated with D2R.

## 2. Results

An autoradiographic analysis of dopamine D2 receptors using [^3^H] methylspiperone radioligand showed its specific binding in many brain regions (Figure 1): the prefrontal cortex (PFc), the cingulate cortex (Cg), the caudate putamen (CPu), the accumbens core (Acc core), the accumbens shell (Acc shell), and the substantia nigra (SN). A statistically significant difference in [^3^H] methylspiperone binding was observed between the WIS and WKY strains in the two regions—i.e., a decrease in the D2 receptor binding in the WKY strain in the PFc; *p* = 0.0007 (Figure 1D), as well as in an increase in SN binding in WKY rats, *p* = 0.0244 (Figure 1I)—when compared to WIS rats.

As a result of the analysis regarding the expression of the gene encoding D2 receptor (*Drd2*), we observed an increase in its expression in the PFc in WKY rats in comparison to WIS rats, *p* < 0.01. Since dopamine D2 receptor signaling can be associated with both canonical and non-canonical pathways, we performed an mRNA expression analysis of the several components involved in intracellular signaling in order to identify the active signaling pathways that are possibly related to the D2 receptor. The results of qRT-PCR analysis indicated that the WKY rat strain had an increased expression of the following genes: *Rgs2*, *p* < 0.05, *Ctnnb1*, *p* < 0.001; *Arrβ2*, *p* < 0.01, *Gsk-3β*, *p* < 0.01; and *Ppp1r1b*, *p* < 0.01 (Figure 2.). No statistically significant changes were observed in the remaining genes under study (*Gnai2*, *p* = 0.2929, *Gnao1*, *p* = 0.9261; *Mapk1*, *p* = 0.7826; and *Grk2*, *p* = 0.6216). (Figure 2).

Since all genes were expressed in the same individuals, we performed a Pearson correlation analysis for the WIS and WKY subjects. The results of this analysis for both strains are presented in the form of a correlation matrix (Figure 3). Firstly, a difference in the correlation between the expression of the *Drd2* dopamine receptor gene and the expression of the genes encoding *Gio* and *Gnai* proteins occurred; however, this correlation was nonsignificant. In the case of the WIS strain, we observed a positive correlation between the *Drd2* and *Gio* expressions (r = 0.61; *p* = 0.08), as well as a negative correlation with the *Gnai* expression (r = −0.60; *p* = 0.086). Conversely, in WKY rats, a positive correlation was observed between the *Drd2* and *Gnai* expressions (r = 0.69; *p* = 0.040). 

Another interesting observation is that although we observed positive correlations in the expression level between the *Arrβ2*, *Ctnnb1*, and *Gsk-3β* genes in both strains of the animals, the correlation map for the two strains looked different. It seems that the key role in WKY rats might be played by the *Rgs2* gene, which showed a strong correlation in expression level with the *Arrβ2* (r = 0.90; *p* < 0.005) and *Ctnnb1* (r = 0.90; *p* < 0.005) genes. Interestingly, in WIS rats, the *Rgs2* gene negatively correlated with the *Drd2* gene expression; however this correlation was nonsignificant *p* = 0.379 (Figure 3). In addition, the correlation map shows that the genes encoding *Grk2* and *Mapk1* are very strongly correlated with each other in the WIS strain (r = 0.94; *p* < 0.005), although no statistically significant changes in the expression of these genes were observed between the WIS and WKY strains (Figure 2).

**Figure 1 ijms-24-05941-f001:**
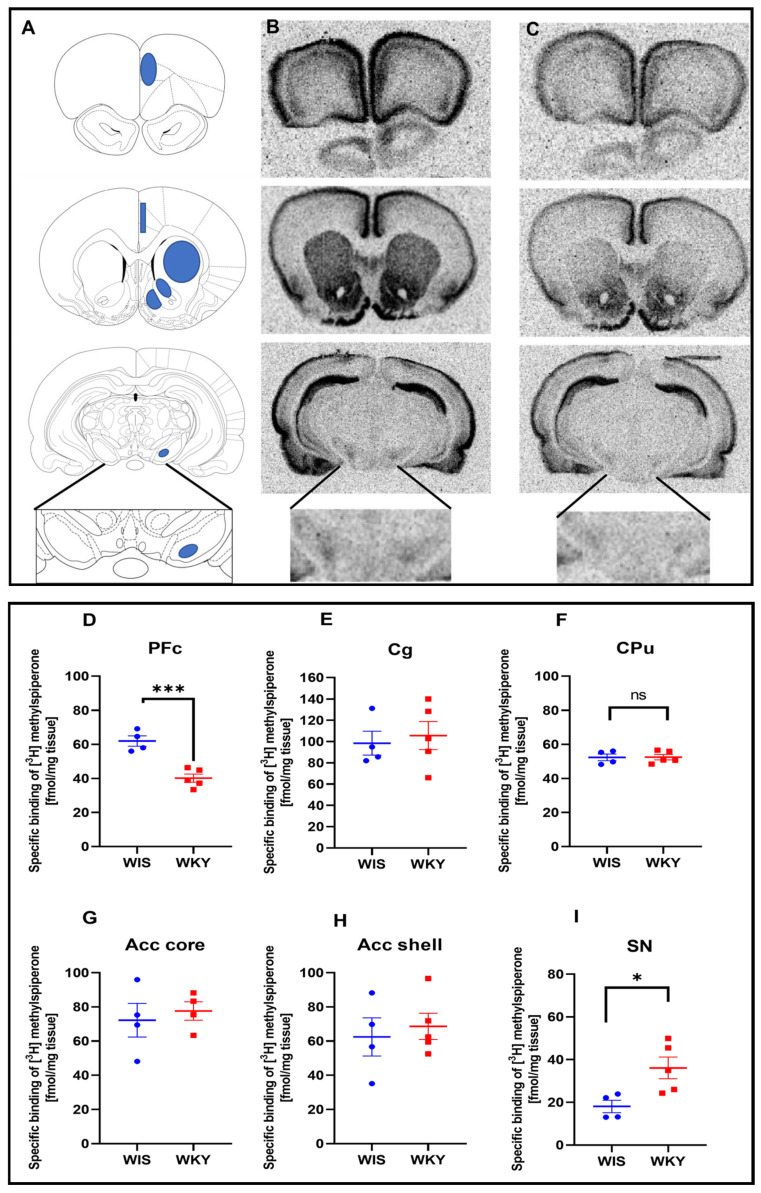
Example autoradiograms of the marked brain areas for which analysis was performed. (**A**) The brain areas were selected for quantitative autoradiography, based on the Rat Brain Atlas [25]. Areas marked in blue were quantified and then converted to fmol/mg tissue. (**B**) The total [^3^H] methyspieperone binding; (**C**) the non-specific binding; (**D**) the decrease in binding in the prefrontal cortex (PFc) (* *p* < 0.05; *** *p* < 0.001 WKY (red points) vs. WIS (blue points)); (**E**) no change in the cingulate cortex (Cg); (**F**) no change (ns) in the caudate putamen (CPu); (**G**) no change in the accumbens core (**H**); no change in the accumbens shell; and (**I**) increase in the binding in the substantia nigra (WKY vs. WIS *p* < 0.05, n = 4).

**Figure 2 ijms-24-05941-f002:**
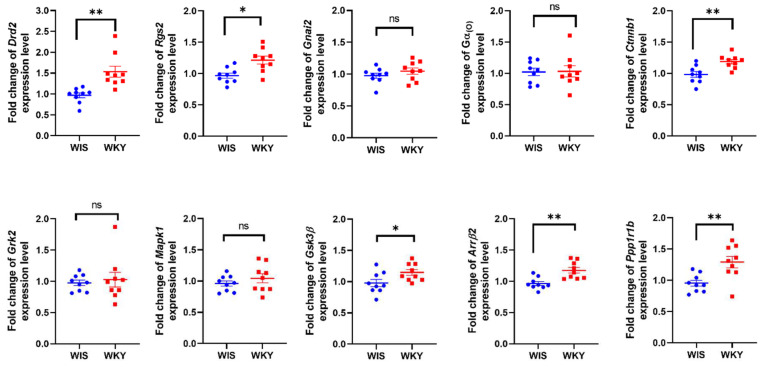
Fold changes in the mRNA expression of the selected genes in the prefrontal cortex (PFc). Individuals are shown in all the figures of n = 9, per group. Statistically significant changes are indicated as * *p* < 0.05, whereas ** *p* < 0.01, ns means no statistically significant changes; WKY (red points) vs. WIS (blue points). Abbreviations: *Drd2*—dopamine receptor D2; *Rgs2*—regulator of G protein signaling 2; *Ctnnb1*—catenin beta 1; *Arrβ2*—arrestin beta 2; *Gsk-3β*—glycogen synthase kinase 3 beta; *Ppp1r1b*—protein phosphatase 1 regulatory inhibitor subunit 1B; *Gnai2*—G protein subunit alpha I2; *Gnao1*—G protein subunit alpha O1; *Mapk1*—mitogen-activated protein kinase 1; and *Grk2*—protein-coupled receptor kinase 2.

**Figure 3 ijms-24-05941-f003:**
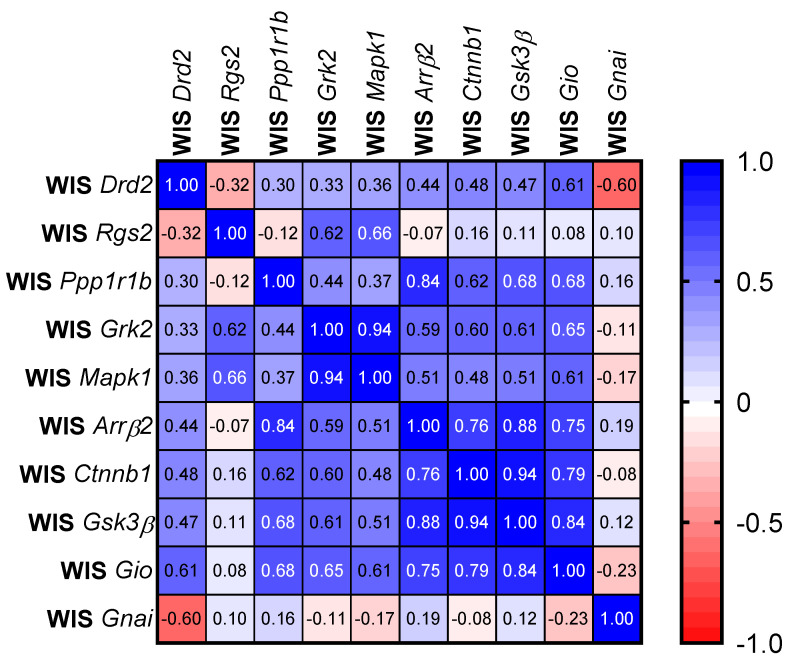
A correlation matrix with Pearson’s r coefficient. Abbreviations: WIS—Wistar Han and WKY—Wistar-Kyoto.

## 3. Discussion

The results of our experiments indicated that several components of the dopamine D2R-receptor-related signaling pathway were significantly different in WKY rats when compared to the WIS strain, which may be responsible for some of the phenotypic features of the WKY rats, including those with treatment-resistant depression.

### 3.1. Rgs2 Is an Important Factor in the Regulation of Dopamine D2 Receptor Activity and Localization

The reduced binding of the radioligand [^3^H] methylspiperone to D2R in the prefrontal cortex of WKY rats may be associated with the increased expression of *Rgs2* in these animals. As previously shown in vitro, RGS2 negatively modulates the D2R-mediated Gαi/o protein signaling in neuroblastoma N2A cells. In addition, RGS2 knockout was shown to abolish quinpirole-stimulated D2R internalization and impaired βarrestin dissociation from the membrane [24].

These data suggest that RGS2 plays an integral role in modulating both D2R activity and localization. The increase in *Rgs2* mRNA expression in WKY rats observed in our study may explain the observed decrease in ligands binding to D2R. RGS proteins act as GTPase-accelerating proteins by facilitating the return of the G protein subunits to the inactive state. By reducing the lifetimes of the G protein-GTP complexes, the RGS proteins act as negative modulators of G protein signaling, and can affect both the potency and the efficacy of the agonist action and of the downstream signaling [19]. In addition, because of the inhibition regarding the activation of the canonical signal transduction pathway via Gi/o proteins, the activation of the non-canonical Arrβ2-related signal transduction pathway may be facilitated. In WKY animals, we observe an increase in the expression of mRNA encoding *Arrβ2*. In addition, the results of the correlation analysis with respect to the expression of individual genes indicated a strong correlation between the expression of *Arrβ2* and *Rgs2*. Moreover, the *Rgs2* gene expression significantly correlated in WKY animals and with the expression of other genes in our study that were related to the βarrestin2/AKT/Gsk-3β/β-catenin signaling pathway. This result confirms that Rgs2 is an important element in the regulation of D2R dopamine receptor activity. Other potential explanations for the decrease in binding to D2R could be that the receptor’s binding site was already occupied by endogenous dopamine (DA). Although there are data indicating an increase in DA in regard to PFc in WKY rats [26], there was a reduction in tissue DA and DOPAC levels, as well as elevations in the DA turnover, which were reported in the WKY PFc, striatum, and/or NAc when compared to WIS rats. These effects were also found to be exacerbated by acute stress [11,15].

### 3.2. The βarrestin2/AKT/Gsk-3β/β-catenin Signaling Pathway

The observed increase in *Arrβ2* expressions in WKY rats may indicate the activation of this specific intracellular pathway. The activation of Arrβ2 leads to an increased expression of *Gsk-3β*, thereby leading to an increased activity of glycogen synthetase 3β (Gsk-3β), which, in turn, phosphorylates β-catenin. It has been shown that dopamine can activate this signal transduction pathway and can be activated independently of the canonical pathway. Moreover, the function of Arrβ2 is important for the expression of dopamine-related behaviors, thereby suggesting that Arrβ2 is a positive mediator of dopaminergic synaptic transmission and a potential pharmacological target for dopamine-related psychiatric disorders [17]. It is interesting to note that it is in WKY rats that we observe the increase in the expression of genes related to this pathway, i.e., *Arrβ2*, *Gsk-3β*, and *Ctnnb1*.

The analyses all indicated that the activated βarrestin2/AKT/Gskβ/β-catenin pathway signaling in PFc is an important characteristic of the WKY strain (Figure 4), especially with respect to the antidepressant drug-resistance phenotype. Gsk-3β is an enzyme that is found in a scaffold protein assembly. These scaffold proteins may undergo specific regulation via different cellular signaling pathways, such as, for example, in βarrestin2/AKT/Gskβ/β-catenin. Preclinical studies have demonstrated the effects of Gsk-3β on mood disorders. It was shown that heterozygous mice with a mutation in the gene encoding *Gskβ* were more resistant to the occurrence of anxiety-depressive behavior [27]. In addition, the intracerebral administration of a Gskβ inhibitor to mice has reduced immobility time in the forced swimming test [28]. It has also been shown that animals carrying mutations for Gsk-3β serine9 have exhibited greater susceptibility to hyperactivity after amphetamine administration, and they have also demonstrated depressive-type behavior in a stressful situation. In a study using the “learned helplessness” model, a decrease in the level of p-ser-Gsk-3β was also observed [29]. Therefore, it seems that modulation at the level of Gsk-3β may be a potential point of drug action in treatment-resistant depression. Of special interest are studies that indicate that one of the most important mechanisms is found in the action of lithium, which is known for its rapid potentiation of the antidepressant effect [30,31]. Lithium is an inhibitor of Gsk-3β activity by the competent inhibition of Mg^2+^ ions, which are required for the activity of Gsk-3. This provides a direct pathway for the inhibition of Gsk-3β by this element [32]. Glycogen synthase kinase is inactivated by serine (ser9) phosphorylation; furthermore, this modification can be catalyzed by several protein kinases, such as PKC (protein kinase C), PKA (protein kinase A), and PKB (protein kinase B). Inactive Gsk-3β can be reactivated by phosphatases that actively remove the phosphate residues. It has been shown that lithium can also inhibit the action of Gsk-3β through an indirect pathway, which involves lithium reducing the action of the phosphatase. This, in turn, leads to the formation of an increased amount of the inactive form of Gsk-3β [33]. Lithium, through a direct inhibition of Gsk-3β, activates the Wnt signaling pathway, which is a complex of glycoproteins involved in embryogenesis, but also in axonal remodeling, the regulation of cell proliferation, stem cell development, and many other processes [34]. In the absence of Wnt, Gsk-3β phosphorylates β-catenins, thus resulting in their rapid degradation. Meanwhile, the binding of Wnt to the frizzled membrane receptor leads to Gsk-3β inhibition and the stabilization of β-catenins, which accumulate in the cell nucleus, where they bind to the T-cell-specific transcription factor (TCF), thereby activating various genes. In the case of WKY rats, we observed an increase in the expression of β-catenins. This is a surprising result, because in the case of chronic stress—which is the main factor in the etiopathogenesis of depression—there is a reduction in phosphorylated Gsk-3β and β-catenins in the PFc, as well as in the administration of the serotonin reuptake inhibitor and citalopram, which restore the levels of these proteins to the control state [35]. Thus, it seems that, in the WKY rats—which are a model of treatment-resistant depression rather than depression generally—there is a different regulation of these genes.

The mechanism of Gsk-3β inhibition may be an important part of the treatment of the drug-resistant pathway, as well as in the increase in the activation of the pathway βarrestin2/AKT/Gsk-3β/β-catenin, which is the cause of non-responsiveness to antidepressants in these animals.

### 3.3. Increased Expression of DARPP 32 in WKY

Of interest is the observed increase in Darpp32 expression. Whether it is a result of the observed decrease in D2R receptor levels requires further investigation. It appears that signaling related to the dopamine D1 receptor may also be responsible for the observed increase in Darpp32 expression. 

The cortex has a relatively high expression level of dopamine receptors [19,36]. However, the detailed expression patterns of D1R and D2R in the PFC are not clear. D1R and D2R have been shown to co-localize in the nucleus accumbens [37,38]. However, it is not clear whether D1R and D2R also co-localize in PFc. The possible co-localization of these receptors may be indicated by electrophysiological studies, which have shown that some cortical neurons respond to both D1R and D2R agonists [39]. Certain mPFC and OFC neurons have also been shown to express both D1R and D2R [40]. 

### 3.4. WKY Strain

The WKY strain was isolated in the 1970s through inbreeding the WIS strain. This was conducted with pressure toward the spontaneous development of hypertension. As research into modeling depressive disorders progressed, it has become apparent that these animals are characterized by several parameters that are identical to the depressive state. It was, therefore, proposed that the WKY strain would be a good genetic animal model of the disorder. These animals exhibit outstanding stress sensitivity and a range of characteristic depressive-type behaviors that are unrelated to stress exposure. This strain differs from others in behavioral, physiological, and neurohormonal responses to environmental, pharmacological, or physical stimuli [8]. For example, when compared to WIS, WKY rats show a hyperreactivity to stress, as measured by open-field tests, hypoactivity in the cross maze, or in the increased immobility that is demonstrated in the forced swim test. They also develop anhedonia more easily in response to stressful stimuli. The mechanism underlying these differences is not clear.

In studies using WKY rats, it is important to compare these animals to WIS rats, which have a similar genetic background [8]. Previous studies comparing the two strains of animals have shown that they differ significantly in terms of biochemistry [9]. Recently, we have demonstrated that several important miRNAs are expressed in the habenula, which differentiated WKY rats from WIS rats and were correlated with changes in the expression of target transcripts [41]. However, it turns out that the WKY strain is more complicated. Recent evaluations of the WKY strains have revealed that not all WKY strains are genetically or behaviorally identical; furthermore, it was found that these characteristics were dependent on the breeding source [42]. The NIH stock of the WKY strain was obtained in 1971 as outbred animals from the Kyoto School of Medicine. These animals were distributed to laboratories such as Harlan and Charles River before the F20 generation, which is the gold standard in obtaining a pure inbred animal [43]. It has recently been shown that two commonly used WKY strains, the WKY/NCrl from Charles River Laboratories and the WKY/NHsd from Harlan Laboratories, display wide genetic divergence [44]. Although this study did not collect any behavioral data, many studies have concluded that the Charles River’s WKY is not a valid control for SHR due to its behavioral abnormalities (i.e., inattention) and is more appropriate as a model of the inattentive-type of attention deficit hyperactivity disorder (ADHD) [44,45,46,47]. Therefore, the WKYs of Charles River origin can serve as a model for inattentive ADHD when compared to the animals of WKY from Harlan Laboratories and the Sprague Dawley rats from the Charles River Laboratories, which have been suggested as controls. This is an interesting observation, especially in the context of the results obtained, as this suggests the involvement of Gsk-3β. It has been shown that the paradoxical sedative effect of amphetamine in hyperactive LAB mice relates to a reduced Gsk-3β activity in the mPFc [48]. Thus, it appears that Gsk-3β may be one of the potential targets of pharmacotherapy for ADHD disorders.

## 4. Materials and Methods

### 4.1. Animals

Male WIS and WKY rats, at the age of 7 weeks and weighing approximately 180–210 g, were acquired from Charles River (Sulzfeld, Germany). Rats were housed in five animals per standard laboratory cage with free access to food and water. Animals were maintained on a 12 h light/dark cycle (lights on at 8 am) under conditions of constant temperature (22  ±  2 °C) and humidity (45  ±  5%). The study was approved by the II Ethical Committee at the Maj Institute of Pharmacology at the Polish Academy of Sciences, Krakow, Poland.

### 4.2. Tissue Preparation

The brains were isolated from decapitated animals. The part of the brain containing the prefrontal cortex was dissected and rapidly frozen on dry ice and stored at −80 °C, until required for further stages. Whole brains were taken from four individuals per group. They were frozen and then sliced, using a Leica cryostat (Jung CM3000, Wetzlar, Germany), into appropriate slices for autoradiographic analysis.

### 4.3. Isolation of mRNAs from the Prefrontal Cortex

The mRNA purification was carried out using an RNAeasy Plus Mini Kit (Qiagen, Hilden, Germany), according to the manufacturer’s instructions. The quantity of the isolated mRNAs was assessed using a NanoDrop ND-1000 (Thermo Fisher Scientific, Waltham, MA, USA), and the quality of the mRNAs was evaluated by microcapillary electrophoresis using an Experion RNA HighSens Analysis Kit (Bio-Rad, Hercules, CA, USA), as per the manufacturer’s recommendations. For further experiments, samples that passed the quality threshold (RIN > 8.0) were used.

### 4.4. Quantitative RT-PCR

The reverse transcription reaction was performed with the High-Capacity cDNA Reverse Transcription Kit (Thermo Fisher Scientific, Waltham, MA, USA), according to the manufacturer’s instructions. The amplified cDNA was diluted with nuclease-free water.

The quantitative PCR reaction was conducted with the Fast SYBR Green Master Mix (Thermo Fisher Scientific, Waltham, MA, USA), following the manufacturer’s recommendations. In addition, it was run, in duplicate, on the CFX96 Touch Real-Time PCR Detection System (Bio-Rad, Hercules, CA, USA). The qPCR reaction was carried out with the following cycles: enzyme activation at 95 °C/3 min, 40 cycles of denaturation at 95 °C/10 s, and then the subsequent annealing/elongation at 60 °C/30 s. The appropriate primer sequences were used (Table 1). Relative gene expression level was calculated with the 2^−ΔΔCT^ method, according to Pfaffl (2001) [49], with a normalization to the reference gene, i.e., the ribosomal protein L32 (*Rpl32*). 

### 4.5. Dopamine D2 Receptor Autoradiography

The D2 receptor autoradiography was performed using the D2R/D3R antagonist [^3^H] methylspiperone (NEN Radiochemicals, Perkin Elmer, Waltham, MA, USA) as the radioactive ligands. Glass slides with 14-µm brain sections were preincubated in 50 mM Tris–HCl, pH 7.4, containing 120 mM NaCl for 15 min at room temperature, followed by a 1-h incubation with [^3^H] methylspiperone (k_d_ = 0.3 nM) in the same buffer containing 0.1 mM ketanserin. Then, the selective serotonin 5-HT2A receptor antagonist (Sigma Aldrich, St Saint Louis, MO, USA) was washed three times/10 min in a Tris-HCl buffer at 0 °C and once/10 min in distilled water at 0 °C. Nonspecific binding was determined on the adjacent slices in the presence of 10 μM (+)butaclamol (Sigma Aldrich, St Saint Louis, MO, USA).

### 4.6. Analysis of the Autoradiographic Images

Finally, the slices after the autoradiography study, were exposed with [^3^H] microscales (GE Healthcare, Pittsburgh, PA, USA) to the tritium-sensitive screens (FujiFilm, Tokyo, Japan) for 7 days. The images were obtained using a FujiFilm BAS 5000 Phosphoimager and analyzed using FujiFilm software (Image Gauge, V4.0, Tokyo, Japan). Specific binding was determined by subtracting non-specific binding from the total binding images. Total and nonspecific binding was performed for each rat. The areas where the analysis was performed were selected based on the Rat Brain Atlas. The same areas were consistently marked for each brain, for both total and nonspecific binding. In addition, the results were expressed as the fmol of the specific radioligand binding per mg protein (fmol/mg) in the brain structures, which were identified according to the Rat Brain Atlas.

### 4.7. Statistical Analysis

The statistical analysis results of all data were compared using an unpaired *t*-test access differentiation between the WIS and WKY rat strains. Pearson’s correlation coefficient was calculated for each pair of the data set, separately, for both the WIS and WKY rats. The correlation data were analyzed using Bonferroni correction (α/10; *p* < 0.005). The analysis was carried out using GraphPad Prism 9.1.2.

## 5. Conclusions

The results obtained in the present study suggest that the WKY strain is characterized by distinct dopamine D2-receptor-dependent intracellular signaling. A key role appears to be played by the RGS2 protein, which, by inhibiting the canonical signal transduction pathway, increases the activity of the non-canonical pathway associated with the βarrestin2/AKT/Gsk-3β/β-catenin pathway. Our data support the role of this pathway in the WKY strain and perhaps explain a potential mechanism of drug resistance in depression associated with Gsk-3β hyperactivity. Despite the observed changes in some intracellular signaling proteins in WKY correlated with dopamine D2Rsignaling, it should be noted that the observed alterations may also involve other G protein-coupled receptors (e.g., dopamine D1 receptor; Figure 4). The data we have presented suggests a correlation between the dopamine D2R and some signaling proteins, but nevertheless in the interpretation of data, the more complicated processes involved in GPCRs signaling must be considered (e.g., GPCRs dimerization [50]).

However, this work has certain limitations due to its methodology. By purifying the material for the study (for both microRNA and mRNA), we were unable to isolate proteins for a Western blot analysis. Certainly, data on phosphorylated proteins would have provided more information that was relevant to the βarrestin2/AKT/Gsk-3β/β-catenin pathway. Nevertheless, the data obtained indicate a role for this pathway in the phenotype of the WKY strain.

## Figures and Tables

**Figure 4 ijms-24-05941-f004:**
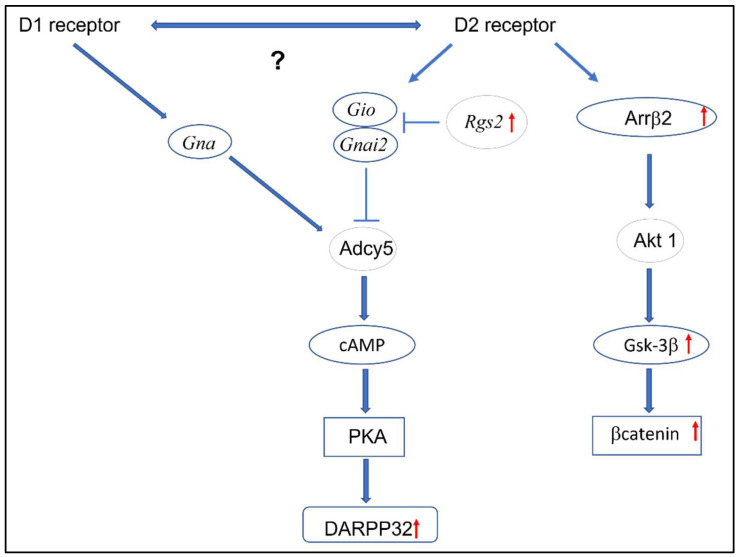
An illustrative diagram showing the signal transduction pathways in the PFc of WKY strain rats.

**Table 1 ijms-24-05941-t001:** Primer sequences.

Gene	Forward	Reverse
*Drd2*	5′AATGGGTCAGAAGGGAA3′	5′AGTGGGCAGGAGATGG3′
*Rgs2*	5′GCGGAGAAAATGAAGCGGACA3′	5′TCTTGCCAGTTTTGGGCTTCCC3′
*Erk2 (Mapk1)*	5′CCTCAAGCCTTCCAACCTC3′	5′GCCCACAGACCAAATATCAATG3′
*Grk2*	5′GCAGCACAAGACCAAAGACA3′	5′CAGTCCAGGGAACGAAAGAA3′
*Arrβ2*	5′ACTTGGACAAAGTGGATCCT3′	5′GGGTCACAAACACTTTCCG3′
*β-catenin (Ctnnb1)*	5′TCCCAGTCCTTCACGCAAGAG3′	5′GTGGCAAGTTCCGCGTCATC3′
*Gsk-3β*	5′AGACCAATAACGCCGCTTCTGC3′	5′AACGTGACCAGTGTTGCTGAGTG3′
*Darp32 (Ppp1r1b)*	5′CATCACTGAAAGCTGTGCA3′	5′TAACTCGTCCTCTTCCTCC3′
*Gnai2*	5′TTCAAGATGTTTGATGTGGG3′	5′GCTATCGAATAGCTTCATGC3′
*Gio (Gnao1)*	5′TTACAAAGGCCAAAGGTCAT3′	5′ AACAAGTTTTTCATCGATAC3′

## Data Availability

The datasets generated and analyzed during the current study are available in the Institute of Pharmacology Polish Academy of Sciences in the Biochemical Pharmacology Laboratory and can be available upon individual request with the corresponding author.

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
