# Peer review of "Altered Intracellular Signaling Associated with Dopamine D2 Receptor in the Prefrontal Cortex in Wistar Kyoto Rats"

_ijms, 2023, doi:10.3390/ijms24065941_

Round 1

Reviewer 1 Report

It is interesting to understand the role of D2 receptors and associated intracellular pathways in the pathology of treatment-resistant depression using an animal model with some face and construct value such as the Wistar Kyoto rat. This paper reports determination of binding for a ligand of D2 receptors and further measured gene expression for various components of G protein-based second messenger mechanisms, leading the authors to conclude that in the prefrontal cortex of W-K rats there is significant change in some of those components that may explain altered transmission at D2 receptors. The experiments and conclusion are generraly sound and reasonable, respectively. However, there are some issues with the analysis of correlation data and other minor issues of interpretation and language use that deserve the attention of the authors as detailed in the following:

-The main issue is that the correlation analysis illustrated in figures 2 and 3 (and explained in the results) presents correlation coefficients without providing information about the statistical significance of those correlations, making it very difficult to estimate how the correlation analysis supports the discussion and conclusions of the authors. Furthermore, any analysis of those correlation data should include a correction for multiple correlations which appears not to have been considered in this report.

-There are some words used for emphasis that are not correctly employed.  For example, the Conclusion section starts with “Undoubtedly” and follows with “suggest”, which clearly calls for removing the word “undoubtedly”. In line 5 of the conclusion “Our data certainly support” should be replaced by simply “Our data support”.

-There are also some grammatical incorrections, such as using “the” instead of “a” in some instances, and other small errors and typos that may be worth revising, maybe with the help of a native or fluent writer of English.

Author Response

Thank you for your valuable comments. Below are our answers.
Manuscript changes are marked in red.
We would like to add that the manuscript has been proofread in MDPI for English editing ID: English-62330.

Reviewer 1

It is interesting to understand the role of D2 receptors and associated intracellular pathways in the pathology of treatment-resistant depression using an animal model with some face and construct value such as the Wistar Kyoto rat. This paper reports determination of binding for a ligand of D2 receptors and further measured gene expression for various components of G protein-based second messenger mechanisms, leading the authors to conclude that in the prefrontal cortex of W-K rats there is significant change in some of those components that may explain altered transmission at D2 receptors. The experiments and conclusion are generraly sound and reasonable, respectively. However, there are some issues with the analysis of correlation data and other minor issues of interpretation and language use that deserve the attention of the authors as detailed in the following:

-The main issue is that the correlation analysis illustrated in figures 2 and 3 (and explained in the results) presents correlation coefficients without providing information about the statistical significance of those correlations, making it very difficult to estimate how the correlation analysis supports the discussion and conclusions of the authors. Furthermore, any analysis of those correlation data should include a correction for multiple correlations which appears not to have been considered in this report.

Thank you for this important and valid comment, Appropriate p-values have been entered into the manuscript. A correction for multiple correlations has also been made. The description has been included in the manuscript.

-There are some words used for emphasis that are not correctly employed.  For example, the Conclusion section starts with “Undoubtedly” and follows with “suggest”, which clearly calls for removing the word “undoubtedly”. In line 5 of the conclusion “Our data certainly support” should be replaced by simply “Our data support”.

It has been changed.

-There are also some grammatical incorrections, such as using “the” instead of “a” in some instances, and other small errors and typos that may be worth revising, maybe with the help of a native or fluent writer of English.

The English correction has been made.

Reviewer 2 Report

The authors describe how the Wistar Kyoto rat, a model of treatment resistant depression, differs from the Wistar Han rat, potentially in its intracellular signaling pathway of the D2-dopamine receptor. The authors conclude that the differences seen could point to why the Wistar Kyoto rat is a good model for depression. 

I am suggesting some edits to help improve the manuscript:

1) On page 2, the authors write about the increased emotionality of the WKY rats. The use of the word emotions is a bit of a stretch when discussing animal models. Perhaps increased anxiety, or something along those lines.

2) Also on page 2, lines 59=60, what are Sst2 and Sst4 receptors? Define the abbreviation.

3) What kind of drug is venlafaxine? (page 2, line 69)

4) The authors keep referring to a Rat Brain Atlas, yet provide no citation or indication of which rat brain atlas it is. 

5) Usually for statistics, you only report the actual p value when it is not significant. For the results presented, I think it would be more useful to write p<0.0001 or whatever that value might be.

6) The graphs in Figure 1 are extremely small and hard to see even when I enlarged them on my computer. 

7) The figure legend for Figure 1 is a little hard to decipher. What type of animals are presented in column B vs column C? It seems like one long run on sentence. Additionally, brain regions do not need to be capitalized. 

8) Figure 2 does not indicate what brain region was used to obtain the PCR data. 

9) On page 4, numerous genes are listed, but it would be helpful to the reader if each of those genes were written out with their full names at least once. 

10) In the Introduction, it says the WKY rats were developed in 1963, while in the discussion it says the 1970s. Please clarify. 

Scientific concerns:

1) Why use autoradiography? It seems slightly out of date and there are improved methods out there to identify the D2 dopamine receptor- antibodies, RNAscope, etc. There needs to be a reason why the authors chose this methodology.

2) There is no mention of Figure 4 in the body of the paper. It should at least be referenced by the authors in the Discussion. 

3) I think it is important to acknowledge that although there were increases in certain intracellular signaling proteins in the WKY rats, that does not necessarily mean that is the only signaling pathway of the D2 receptor. Numerous studies have examined the signaling pathway of the D2 receptor, so I am interested to see how the authors would reconcile what is known about the D2 receptor with what they found in the WKY rats. Remember correlation is not causation. 

Author Response

Thank you for your valuable comments. Below are our answers. Manuscript changes are marked in red. We would like to add that the manuscript has been proofread in MDPI for English editing ID: English-62330.

The authors describe how the Wistar Kyoto rat, a model of treatment resistant depression, differs from the Wistar Han rat, potentially in its intracellular signaling pathway of the D2-dopamine receptor. The authors conclude that the differences seen could point to why the Wistar Kyoto rat is a good model for depression. 

I am suggesting some edits to help improve the manuscript:

  • On page 2, the authors write about the increased emotionality of the WKY rats. The use of the word emotions is a bit of a stretch when discussing animal models. Perhaps increased anxiety, or something along those lines.

Thank you for your suggestion. It has been changed.

  • Also on page 2, lines 59=60, what are Sst2 and Sst4 receptors? Define the abbreviation.

The abbreviation has been defined.

  • What kind of drug is venlafaxine? (page 2, line 69)

Venlafaxine is an antidepressant belonging to serotonin and noradrenaline reuptake inhibitors (SNRIs). This information was added to the manuscript.

  • The authors keep referring to a Rat Brain Atlas yet provide no citation or indication of which rat brain atlas it is. 

The citation has been added.

  • Usually for statistics, you only report the actual p value when it is not significant. For the results presented, I think it would be more useful to write p<0.0001 or whatever that value might be.

It has been changed.

  • The graphs in Figure 1 are extremely small and hard to see even when I enlarged them on my computer. 

The quality of the graph has been changed.

  • The figure legend for Figure 1 is a little hard to decipher. What type of animals are presented in column B vs column C? It seems like one long run on sentence. Additionally, brain regions do not need to be capitalized. 

It has been changed. Column B represents the autoradiogram of Total binding, column C represents Nonspecific binding for the same rat.

8) Figure 2 does not indicate what brain region was used to obtain the PCR data. 

A relevant note has been included in the description of the figure.

9) On page 4, numerous genes are listed, but it would be helpful to the reader if each of those genes were written out with their full names at least once. 

It has been added.

10) In the Introduction, it says the WKY rats were developed in 1963, while in the discussion it says the 1970s. Please clarify. 

The WKY rats are developed from an outbred Wistar stock from the Kyoto School of Medicine to NIH 1971. The date has been changed in the introduction.

Scientific concerns:

  • Why use autoradiography? It seems slightly out of date and there are improved methods out there to identify the D2 dopamine receptor- antibodies, RNAscope, etc. There needs to be a reason why the authors chose this methodology.

In our laboratory, we routinely apply binding analyses using radioligands. The object of our research interest also includes the dopamine D2 receptor. Having had numerous experiences using antibodies to this receptor, we believe that a more selective and specific method is an analysis using a radioligand. The method has high resolution and the use of specific radioligands is the best technique for studying membrane receptors. Antibodies are not always specific and often give false-positive results. In addition, immunohistochemistry is not a good method for quantitative comparisons and autoradiography with an appropriate scale is quantitative.

  • There is no mention of Figure 4 in the body of the paper. It should at least be referenced by the authors in the Discussion. 

It has been added.

3) I think it is important to acknowledge that although there were increases in certain intracellular signaling proteins in the WKY rats, that does not necessarily mean that is the only signaling pathway of the D2 receptor. Numerous studies have examined the signaling pathway of the D2 receptor, so I am interested to see how the authors would reconcile what is known about the D2 receptor with what they found in the WKY rats. Remember correlation is not causation. 

Thank you for this comment. We agree with the reviewer. The relevant paragraph has been added in the Conclusion section.

Reviewer 3 Report

The manuscript of Agata Korlatowicz et al., entitled „Altered intracellular signaling associated with dopamine D2 receptor in the prefrontal cortex in Wistar Kyoto rats” reveals new data about dopamine receptor signaling abnormalities in Wistar Kyoto rats. The results can be interesting for experts in the field, but it needs revision before publication.

 1. It should be clarified, whether methylspiperone is a D2 receptor specific agent, or rather a D2-like receptorfamily (D2/D3) specific agent.

 2. If WKY abbreviation occurs in the abstract, it should be explained in the abstract as well.

 3. The following sentence seems to be too general: „depression is related to a decrease in the sensitivity of postsynaptic D2 dopa-65 mine receptors (D2R) in the brain areas that are responsible for reward”, and it is probably not valid for all brain regions that are involved in reward circuits.

 4. Abbreviations should be introduced, e.g. Galpha, DARPP, etc.

 5. At least one sentence about WIS Han rats should be included in the introduction. Why this strain is the control?

 6. Quality of Figure 1 should be significantly improved, e.g. D-I could be below A-C and not next to them. It should be also explained more detailed in the methods, how the borders of the selected areas were identified, and how was it taken into consideration, when the binding was inhomogenous, as seems to be in the most cases in Fig. 1.

 7. In line 131, it is not clear, that the negative correlation occurs between Drd2 and Gnai, it could mean between Gnai and Gio as well.

 8. If WKY and WIS abbreviations were already introduced, both of them could be applied in materials and methods as well.

 9. Revision of grammar and spelling is necessary, e.g. ) is missing in line 59-60, line 61-62 alo would need a revision („Besides other monoaminergic systems, mesolimbic and mesocortical dopamine function is decreased in WKY rats [10-15] may also contribute”), is and may seems to be too much together in the same sentence.

Author Response

Thank you for your valuable comments. Below are our answers.
Manuscript changes are marked in red.
We would like to add that the manuscript has been proofread in MDPI for English editing ID: English-62330.

The manuscript of Agata Korlatowicz et al., entitled „Altered intracellular signaling associated with dopamine D2 receptor in the prefrontal cortex in Wistar Kyoto rats” reveals new data about dopamine receptor signaling abnormalities in Wistar Kyoto rats. The results can be interesting for experts in the field, but it needs revision before publication.

  1. It should be clarified, whether methylspiperone is a D2 receptor specific agent, or rather a D2-like receptorfamily (D2/D3) specific agent.

Thank you for this comment. The description has been changed. The methylspiperone is D2/D3 specific radioligand. However, D3 receptors appear to be more associated with the basal ganglia and structures involved in motor behavior, while D2 was associated with regions related to cognitive/affective functions.

  1. If WKY abbreviation occurs in the abstract, it should be explained in the abstract as well.

It has been added.

  1. The following sentence seems to be too general: „depression is related to a decrease in the sensitivity of postsynaptic D2 dopa-65 mine receptors (D2R) in the brain areas that are responsible for reward”, and it is probably not valid for all brain regions that are involved in reward circuits.

Thank you for this suggestion. It has been changed.

  1. Abbreviations should be introduced, e.g. Galpha, DARPP, etc.

It has been added.

  1. At least one sentence about WIS Han rats should be included in the introduction. Why this strain is the control?

It has been explained in the introduction.

  1. Quality of Figure 1 should be significantly improved, e.g. D-I could be below A-C and not next to them. It should be also explained more detailed in the methods, how the borders of the selected areas were identified, and how was it taken into consideration, when the binding was inhomogenous, as seems to be in the most cases in Fig. 1.

The quality of the graph has been changed. Total and nonspecific binding was performed for each rat. The areas where the analysis was performed were selected based on the rat brain atlas. The same areas were consistently marked for each brain, for both total and nonspecific binding.

  1. In line 131, it is not clear, that the negative correlation occurs between Drd2 and Gnai, it could mean between Gnai and Gio as well.

The p-value indicates that the correlation between Gnai and Gio is not significant.

  1. If WKY and WIS abbreviations were already introduced, both of them could be applied in materials and methods as well.

It has been changed.

  1. Revision of grammar and spelling is necessary, e.g. ) is missing in line 59-60, line 61-62 alo would need a revision („Besides other monoaminergic systems, mesolimbic and mesocortical dopamine function is decreased in WKY rats [10-15] may also contribute”), is and may seems to be too much together in the same sentence.

The English correction has been made.

Round 2

Reviewer 3 Report

The manuscript has been improved significantly.